# Mind the gap: A method for evaluating and comparing regional knowledge in LLMs

## Abstract

Large Language Models (LLMs) achieve strong results on general knowledge benchmarks, yet their coverage of region-specific entities—particularly from Latin America—remains limited. To address this gap, we propose CHOCLO, an entity-centric methodology for evaluating LLM knowledge of culturally relevant entities in Latin America. The methodology extracts structured facts from domain-specific resources and organizes them into knowledge graphs spanning seven categories, resulting in more than 44,000 entities and 130,000 questions. Evaluation is carried out through two complementary strategies. The first computes factual scores using LLM-as-a-judge as a measure of accuracy. The second trains probing models that predict these scores directly from LLM embeddings, enabling generation-free evaluation. Results consistently show a regional disparity: GPT-5 and GPT-3.5 score markedly lower on Latin American entities compared to the U.S. and Europe, while models such as Mistral, DeepSeek, and QWEN underperform across all regions. Category-level analysis further reveals that fauna, flora, and traditions are comparatively better represented, whereas public figures and objects show the largest deficits. CHOCLO thus exposes systematic disparities in how LLMs encode Latin American knowledge and provides a step toward culturally inclusive benchmarks that support fairer global evaluation.

## 1 Introduction

*"In diversity there is beauty and there is strength."*
— Maya Angelou

Large Language Models (LLMs) have achieved impressive performance across a wide range of tasks, including reasoning, mathematical problem solving, instruction following, and code generation (Arora et al., 2023; Wang et al., 2023; Yang et al., 2025; Li, 2024; Sun et al., 2024; Hossain et al., 2025). However, their factual knowledge remains unevenly distributed, reflecting geographic and cultural biases inherent in the web-scale corpora used during pretraining. These datasets disproportionately represent high-resource regions and data-rich languages, while content from underrepresented areas, such as Latin America (Latam), receives only limited coverage.

Evaluating LLMs only on widely known or commonly represented facts is similar to testing students exclusively on memorized, surface-level questions: most will succeed, but such evaluation provides little insight into deeper comprehension. For example, while LLMs can confidently recall stereotypical encyclopedic knowledge about salient geographic entities from Latam, such as *Cataratas del Iguazú* or *Buenos Aires*, they often struggle with situated knowledge that is culturally and locally relevant, such as *Carlos Caszely* (a famous Chilean soccer player) or *Chipa* (a traditional Paraguayan bread). These cases highlight the need to measure knowledge beyond the high-frequency entities that dominate pretraining corpora.

Most existing benchmarks for evaluating the cultural and regional knowledge of LLMs rely on question-answering (QA) tasks (Myung et al., 2024; Chiu et al., 2024; Dadas et al., 2025; Shen et al., 2024; AlKhamissi et al., 2024; Romero et al., 2024; Grandury et al., 2025). Although QA metrics capture model correctness, they treat facts in isolation and provide little insight into how much a model knows about a particular entity as a whole.

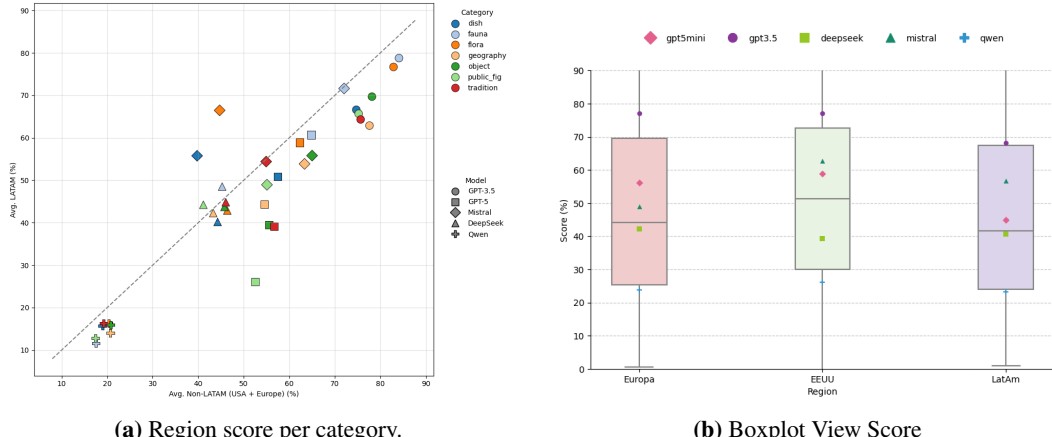

**(a)** Region score per category.       **(b)** Boxplot View Score

**Figure 1:** Comparison of language model performance across regions and categories. (a) Relationship between performance in Latin America and non-LATAM regions (USA + Europe), broken down by entity category (dish, fauna, flora, geography, object, public figure, tradition). Each point corresponds to a model in a specific category; symbols distinguish models and colors represent categories. The diagonal indicates equal performance between LATAM and non-LATAM. (b) Distribution of average llm-as-judge scores for five models (gpt5mini, gpt3.5, DeepSeek, Mistral, and Qwen) across three geographic regions (Europe, USA, and Latin America). Boxplots show variability within each region, while markers indicate the average score achieved by each model.

As shown in Figure 1, even the strongest LLMs exhibit consistent performance gaps between Latin America and the US/Europe, with the most significant drops occurring in socially grounded categories such as public figures and objects. Interestingly, Mistral shows comparatively stronger performance in several LATAM categories, often reducing the gap relative to GPT-3.5 and GPT-5, yet it remains well below its own Non-LATAM scores. This indicates that the regional disparity persists across all models, even for those that handle LATAM content slightly better.

Knowledge graphs provide a systematic representation of the complete knowledge associated with an entity, analogous to a comprehensive exam that evaluates all aspects of a subject. By structuring information into triplets ⟨*entity, relation, value*⟩ (Hogan et al., 2021), they enable fine-grained assessment of how well different models capture entity-level knowledge. This structured representation is beneficial for identifying gaps between regions and comparing performance across semantic categories, offering a consistent basis for cross-regional evaluation.

In this work, we propose *CHOCLO*—named after maize, the foundation of food, culture, and identity in Latin America—a novel methodology for evaluating the knowledge of LLMs about entities across domains such as traditions, public figures, food, and geography. Unlike QA-based benchmarks (Chiu et al., 2024; Myung et al., 2024), *CHOCLO* adopts an entity-centric perspective, assessing whether models can reconstruct the structured relations associated with each entity.

Our methodology combines two complementary strategies. First, we **compute factual scores** comparing model-generated relations against a ground-truth knowledge graph using LLM-as-judge evaluation accuracy. Second, we introduce a **probing model** that takes advantage of the LLM's own embeddings to predict knowledge scores without requiring text generation, allowing an efficient and generalizable evaluation on unseen entities.

Our contributions are the following:

1. We introduce **CHOCLO**, a benchmark of culturally relevant entities from underrepresented regions, such as Latin America, organized into seven thematic domains and connected through a structured knowledge graph.

2. We contribute an **entity-centric evaluation method** that measures factual knowledge of LLM using assessment of LLM as a judge accuracy.

3. We will release a **probing model** that predicts these scores directly from LLM embeddings, enabling efficient, generation-free evaluation of unseen entities and serving as a test of the generalizability of our methodology.

4. We will incorporate a cultural and regional perspective into the evaluation of LLM, positioning Latin America as a central axis. CHOCLO extends the evaluation to Latin American knowledge by generating structured questions grounded in culturally relevant entities.

The remainder of this paper is structured as follows: Section 2 reviews related work. Section 3 details the dataset construction and methodology. Section 4 presents experimental results across LLMs, categories, and regions. Section 5 concludes with directions for future research.

## 2 RELATED WORK

Several recent studies have introduced benchmarks to evaluate cultural and regional knowledge in LLMs (Myung et al., 2024; Chiu et al., 2024; Dadas et al., 2025; Shen et al., 2024; AlKhamissi et al., 2024; Adebara et al., 2025). However, most efforts focus on question-answering accuracy or language-specific tasks, often overlooking structured evaluations of factual knowledge at the *entity level*, particularly for underrepresented regions such as Latin America. Despite growing awareness of regional disparities in model performance, there has been limited attention to systematic assessments of culturally grounded knowledge using structured resources such as knowledge graphs.

Gottesman & Geva (2024) proposed KEEN a probing method that estimates how much an LLM knows about a given entity by training an MLP regressor on internal embeddings and a QA-derived knowledge score. While our work adopts a similar probing approach, it diverges in two key ways. First, we compute ground-truth scores directly from the LLM's ability to reconstruct factual knowledge graph connections, rather than relying solely on QA correctness. Second, we explicitly target culturally significant Latin American entities, a region often marginalized in prior evaluation benchmarks. Furthermore, we also expand the evaluation space by proposing four complementary scoring methods: lexical token overlap, embedding similarity to the ground truth, multiple-choice factual verification, and LLM-as-a-judge validation.

Benchmarks such as CulturalBench (Chiu et al., 2024) and BLEnD (Myung et al., 2024) explore regional and linguistic disparities through curated QA datasets across the US, Europe, and the Global South. Although valuable for quantifying high-level geographic variation, they lack the structural granularity to assess entity-level compositional knowledge. Other region-specific benchmarks take multimodal or linguistic perspectives. CVQA (Romero et al., 2024) evaluates cultural diversity in vision-language models through image-based QA, and La Leaderboard (Grandury et al., 2025) tests generative LLMs in Spanish dialects and Latin American varieties. Although these resources address representation gaps, they remain primarily focused on linguistic or generative capabilities rather than factual world knowledge encoded at the entity level.

Adebara et al. (2025) introduce a broad benchmark for African languages spanning classification, generation, and reasoning tasks. Their findings highlight performance disparities in low-resource settings, but again without evaluating structured factual knowledge. In contrast, our methodology focuses on how well models reproduce culturally grounded knowledge graphs for specific entities, offering a new dimension of evaluation that captures depth of knowledge beyond surface-level correctness.

Finally, ToolHop (Ye et al., 2025) exemplifies the trend toward structured, interpretable evaluations by focusing on the use and reasoning of multihop tools. Although not culturally focused, it emphasizes task-based interpretability, an objective we share. Our methodology extends this idea by modeling factual knowledge as structured entity graphs, allowing principled analyses of model coverage, gaps, and hallucinations.

In summary, compared to existing benchmarks, our methodology advances entity-centric evaluation in two ways: (1) by introducing multiple complementary scoring methods –lexical, semantic, judgment-based, and multiple choice– that jointly capture different facets of factual knowledge; and (2) by combining these factual scores with regression-based probing to assess generalization to unseen entities. This dual perspective enables us to transition from surface-level QA benchmarks to a

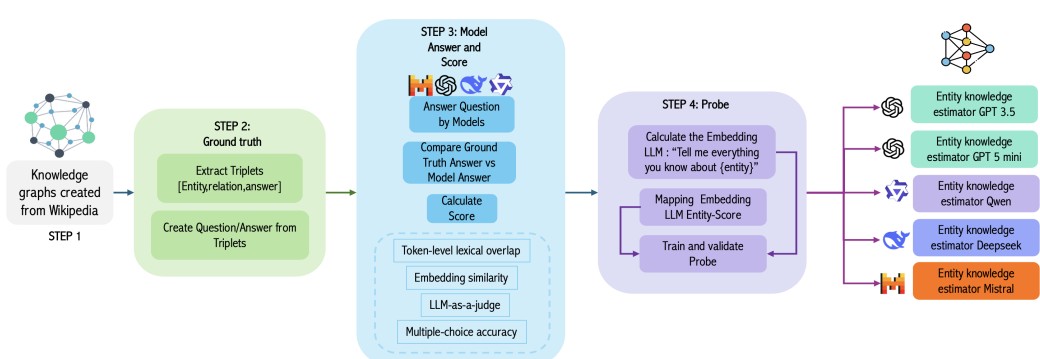

**Figure 2:** Overview of CHOCLO. Entities are selected by category and region, linked to knowledge graph triplets, and used to generate evaluation questions. LLM responses are scored with four metrics (lexical, embedding similarity, LLM-as-judge, and multiple-choice matching) plus an average. Entity embeddings paired with scores are then used to train an MLP regressor, enabling evaluation of generalization to unseen entities across regions.

deeper, structured understanding of LLM knowledge regarding culturally significant entities, with a particular focus on Latin America.

## 3 METHODOLOGY

In this section, we describe the methodology underlying our proposed *CHOCLO* method. We begin by introducing the dataset construction process, including entity selection, category design, and regional coverage (Tables 1 and 4). We then present the overall *CHOCLO* pipeline (Figure 2), which integrates knowledge graph extraction and the generation of question–answer (QA) pairs. These QA pairs are not reported as a standalone benchmark; instead, they provide supervision signals used to train a probe-based classifier.

The core contribution of *CHOCLO* lies in this probe-based evaluation: given an entity-specific embedding obtained from a model (via prompts such as *"Tell me everything you know about entity"*), the classifier predicts a knowledge score calibrated from QA-derived labels. This allows us to estimate how consistently different LLMs represent regional and cultural knowledge, without relying on handcrafted templates. A note of clarification is that the predicted score reflects the calibration ability of the probe, not the absolute amount of knowledge stored by the model.

### 3.1 DATA COLLECTION AND ENTITY COVERAGE

To evaluate the knowledge of culturally meaningful entities in LLMs, we construct a structured data set based on factual descriptions retrieved from the *Wikidata Cultural Entity API*. We focus on three regions –Latin America, Europe, and the United States– and select entities across seven semantic categories adapted from CVQA(Romero et al., 2024): *Dish*, *Fauna*, *Flora*, *Geography*, *Object*, *Public Figure*, and *Tradition*. This design ensures broad thematic coverage while capturing culturally salient patterns. Although entities retrieved from Wikidata can often be linked to multiple thematic domains or geographic areas, in our dataset each entity is annotated with exactly one semantic category and one region, except for dishes, which may be associated with more than one region due to significant differences in preparation and local variations. For category assignment, we map Wikidata descriptions and properties onto our seven adapted axes, selecting the most representative category when more than one was plausible. For each category and region, we selected entities that are highly representative of their cultural context, ensuring balanced coverage across categories while emphasizing cultural salience.

For regional assignment, we apply a salience rule that prioritizes structured Wikidata signals—such as country of origin (property P495), country of citizenship (P27), place of birth (P19), territorial

location (P131), and geographic coordinates (P625)—augmented by lead descriptions in Spanish/-Portuguese when needed. These identifiers (commonly referred to as P-codes) are standardized properties in Wikidata that explicitly link entities to geographic or cultural attributes. The final label is always the macro-region (Latin America, Europe, or United States), with country-level anchors retained only as supporting evidence. Examples within Latin America illustrate this process: *Carlos Caszely* is assigned to Latin America with Chile as the anchor, since both citizenship (P27) and football career consistently point to Chile; *Charrúa* is linked to Latin America with Uruguay as the anchor, because ethnographic descriptions and historical territory (P131) converge in present-day Uruguay; and *Chipa* is annotated under Latin America with Paraguay as the primary anchor (country of origin, P495), while still acknowledging documented variants in northern Argentina. These examples show how our procedure ensures consistency across annotations while preserving cultural specificity.

**Table 1:** Unique entities per category across regions; Number of entities (**#Ents**) and total questions (**#Qs**).

| Category | LATAM | | EUROPE | | USA | | Total | |
|---|---|---|---|---|---|---|---|---|
| | **#Ents** | **#Qs** | **#Ents** | **#Qs** | **#Ents** | **#Qs** | **#Ents** | **#Qs** |
| Dish | 1,942 | 5,826 | 2,273 | 6,819 | 777 | 2,331 | 4,992 | 14,976 |
| Fauna | 2,444 | 7,332 | 2,604 | 7,812 | 2,652 | 7,956 | 7,700 | 23,100 |
| Flora | 2,075 | 6,225 | 1,762 | 5,286 | 1,512 | 4,536 | 5,349 | 16,047 |
| Geography | 2,299 | 6,897 | 2,431 | 7,293 | 2,517 | 7,551 | 7,247 | 21,741 |
| Object | 738 | 2,214 | 2,220 | 6,660 | 2,079 | 6,237 | 5,037 | 15,111 |
| Public Figure | 2,774 | 8,322 | 2,578 | 7,734 | 2,524 | 7,572 | 7,876 | 23,628 |
| Tradition | 1,979 | 5,937 | 2,336 | 7,008 | 2,141 | 6,423 | 6,456 | 19,368 |
| **Tot.** | 14,251 | 42,753 | 16,204 | 48,612 | 14,202 | 42,606 | 44,657 | 133,971 |

Table 1 details the distribution of entities and questions per category and region, totaling 44,657 entities and 133,971 questions. Compared to previous benchmarks such as CulturalBench (Chiu et al., 2024) (1,696 questions) or BLEnD (Myung et al., 2024) (52k questions), our dataset provides substantially broader coverage, with balanced representation across regions and categories.

To assess potential biases inherited from the source data, we examined both the total number of entities extracted from Wikidata per region (Figure 3a) and the proportion of those entities that have in the Web (Figure 3). Although Latin America, the United States, and Europe yield comparable numbers of entities at extraction time, the coverage analysis reveals a clear and persistent asymmetry: Europe and the U.S. show markedly higher web presence across all categories, whereas Latin American entities remain systematically underrepresented. This pattern reflects well-documented regional coverage disparities in Wikidata and Web, confirming that the imbalance stems from the structure of the underlying knowledge sources rather than from our benchmark design. Furthermore, despite being trained on corpora far broader than web-search alone, current LLMs do not compensate for this discrepancy, as their knowledge of Latin American entities remains comparatively weaker. Together, Figure 3a and Figure 3 highlight the need for a benchmark like CHOCLO, which explicitly evaluates model performance in regions that have historically been underdocumented.

Entities are divided into training, validation, and test sets (see Appendix A.2), allowing controlled experiments on both in-domain and unseen entities. Each entity is linked to a set of ground-truth triplets of the form $\langle subject, relation, object \rangle$, filtered for relevance and deduplicated to form concise knowledge graphs. To ensure robust evaluation of probe models, we adopt a 5-fold cross-validation setting, reporting the mean and standard deviation of RMSE across folds. This setup reduces variance due to split choice and provides a more reliable estimate of model generalization across regions and categories. These triplets are later transformed into natural language questions using templated prompts, ensuring consistency across categories. An illustration of this process is shown as follows, where a knowledge-graph fact is mapped into a question-answer pair through a relation-based template:

Figure 5 shows a cross-country variation in model performance. Concrete and perceptually grounded categories such as dish, flora, fauna tend to generalize more reliably across the region, whereas culturally specific and socially anchored categories, particularly *public figure*—display consistently

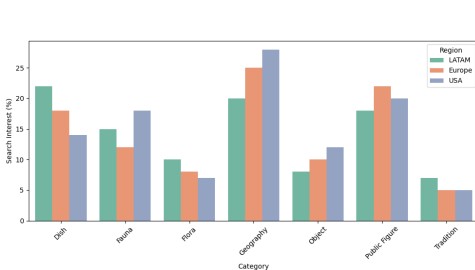 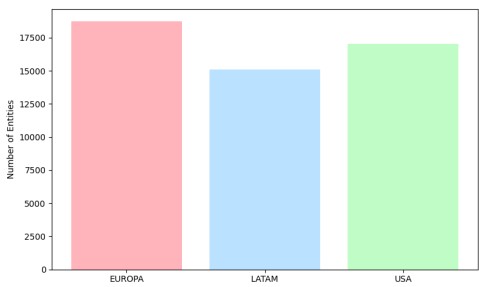

**(a)** Total number of entities extracted from Wikipedia web search per region (LATAM, USA, Europe).

**(b)** Proportion of entities with an existing Wikipedia web search, grouped by category and region.

**Figure 3:** Combined analysis of Wikidata entity extraction volume (left) and web coverage rates (right).

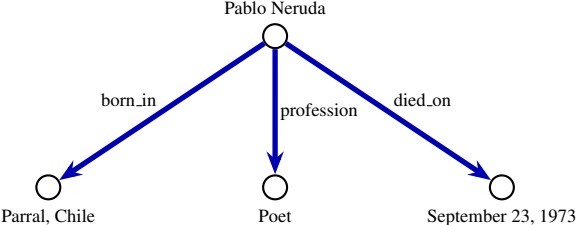

> **Example (KG triplet):** ⟨*Pablo Neruda, profession, Poet*⟩
> **Template:** *"What is the {relation} of {subject}?"*
>
> **Generated Question:** *What is the profession of Pablo Neruda?*
> **Expected Answer:** *Poet*

**Figure 4:** Example of question generation from a knowledge graph (KG) triplet. The triplet ⟨Pablo Neruda, profession, Poet⟩ is converted into a natural language question using a template ("What is the relation of subject?"), yielding the generated question "What is the profession of Pablo Neruda?" with the expected answer Poet.

low scores. This pattern suggests that current LLMs retain broad factual or object-level knowledge but struggle with fine-grained, localized cultural information in Latin America.

As shown in Figure 6, both evaluation frameworks—CulturalBench (Chiu et al., 2024) and our LLM-as-judge benchmark—reveal a clear and consistent performance gap between Latin America (LATAM) and the other regions when evaluating GPT-3.5. CulturalBench reports an average accuracy of 0.741 for LATAM, compared with 0.771 for Europe and 0.825 for the United States. Our relevance-based metric exhibits an even sharper separation: LATAM reaches 0.693, whereas Europe and the United States obtain 0.782 and 0.787, respectively.

## 3.2 PROPOSED METHOD

Figure 2 presents our proposed method. Entities are selected by category and region, mapped to knowledge graph triplets, and transformed into question–answer (QA) pairs as show in Figure 3. These QA-derived labels are then used to train a probe-based classifier, which takes entity-specific embeddings from each LLM and predicts a corresponding knowledge score. This design empha-

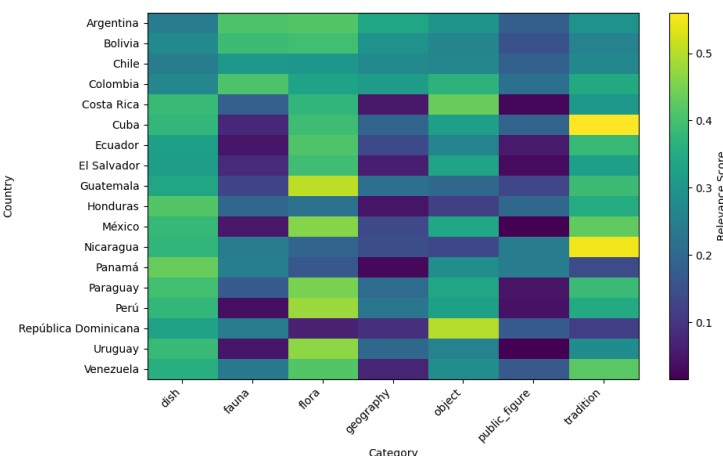

**Figure 5:** Heatmap of average LLM-as-judge score across Latin American countries and CHOCLO categories.

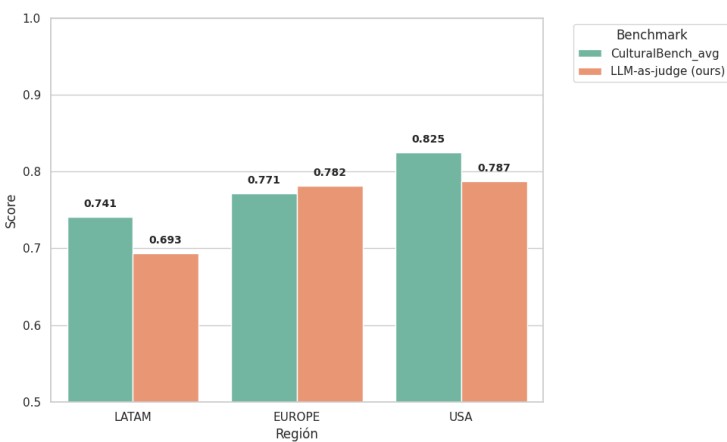

**Figure 6:** Comparison between CulturalBench (avg easy+hard) and our LLM-as-judge benchmark across regions.

sizes the classifier as the core contribution, showing how knowledge can be estimated directly from embeddings while supporting interpretable error analysis.

### 3.3 Human Validation and Agreement Levels

Approximately 67% of the benchmark answers, specifically those below 60% in the LLM-as-judge score on our best-performing model (GPT-3.5), were manually reviewed by human experts. Each question was inspected once in a single-pass validation.

**Table 2:** Expert agreement rates across regions and cultural categories.

| Region | Dish | Fauna | Flora | Geography | Object | Public Fig. | Tradition | Avg |
|--------|------|-------|-------|-----------|--------|-------------|-----------|-----|
| LATAM | 87.5% | 87.5% | 83.3% | 75.0% | 85.71% | 85.71% | 83.3% | **84.0%** |
| Europe | 90.0% | 87.5% | 88.8% | 85.7% | 88.8% | 88.0% | 88.9% | **87.1%** |
| USA | 88.9% | 88.9% | 85.7% | 88.9% | 85.7% | 85.7% | 90% | **87.9%** |

Table 2 shows that all categories achieve at least 75% agreement, and most exceed 85%. Europe and the United States display slightly higher consistency, reflecting more standardized documentation,

but Latin America also reaches strong agreement across all domains. These results confirm that the validated subset provides a stable reference point for evaluating the LLM-as-judge metric.

### 3.3.1 GRAPH-BASED EVALUATION

Each ground-truth triplet $\langle s, r, o \rangle$ is converted into a natural language question, and the LLM's predicted answer $\hat{o}$ is scored under **LLM-as-a-judge** which considers binary semantic equivalence decision from a reference LLM.

For each entity, the final knowledge score is the average across its set of questions:

$$K(e) = \frac{1}{m} \sum_{j=1}^{m} s(o_j, \hat{o}_j),$$

### 3.3.2 PROBE-BASED EVALUATION

To complement our graph-based scoring, we probe LLM embeddings directly. Following the KEEN (Gottesman & Geva, 2024) methodology, for each entity $e$, the evaluated LLM is prompted with *"Tell me everything you know about {entity}"*. We extract the resulting hidden representation $h(e) \in \mathbb{R}^d$, which serves as input to a lightweight MLP regressor:

$$\hat{K}_{\text{probe}}(e) = f_\theta(h(e)),$$

where $f_\theta$ is trained to predict graph-based knowledge scores. The architectural details of the MLP regressor are presented in Appendix A.1, while Appendix A.4 illustrates example questions derived from Knowledge Graph triplets, along with the scores obtained by different models.

### 3.4 EVALUATION SETUP

We evaluate five LLMs—GPT-3.5 Turbo, GPT-5 Mini, DeepSeek-llm-7b-chat, Qwen1.5-0.5B, and Mistral 3.1 Small (24B)—across the full dataset. Each model is used in two complementary roles. First, they generate answers that are evaluated using an LLM-as-a-judge evaluation metric. Second, they produce entity embeddings in response to a canonical prompt of the form *"Tell me everything you know about {entity}"*, which serve as inputs for the probe-based evaluation. For OpenAI embeddings (GPT-3.5 and GPT-5), we compute both the large and small versions. Each embedding version is then trained with supervision from both GPT-3.5 and GPT-5 factual scores, yielding a total of four probe configurations (small+GPT-3.5, small+GPT-5, large+GPT-3.5, and large+GPT-5). In this way, the same models provide both the outputs for direct scoring and the internal representations for probing.

To assess generalization, the entities are divided into train, validation, and test sets (Table 4). Probe models are trained on the training set and evaluated using a 5-fold cross-validation scheme, ensuring that prediction errors reflect performance on unseen entities. Finally, we report regional average scores (Table 3) and category-level performance (Table 5). This two-way perspective highlights both geographic disparities and domain-specific gaps, enabling a precise assessment of how much LLMs know about culturally relevant entities in Latin America compared to those in Europe and the United States.

## 4 EXPERIMENTS

In this section, we present the experimental results obtained with our method. Question–answer pairs derived from the knowledge graph provide supervision signals that enable training a probe-based classifier, which predicts knowledge scores directly from entity embeddings. Results are reported at two levels: (i) by region (Latin America, Europe, and the United States), capturing differences in how consistently the probe estimates knowledge across geographic contexts, and (ii) by category, highlighting thematic domains where estimation is more reliable or challenging. The experiments also evaluate the generalization capability of probe models trained on entity embeddings under a cross-validation setup. Taken together, these analyses demonstrate how LLM representations encode culturally grounded knowledge, with scores reflecting the calibration ability of the probe rather than absolute knowledge levels.

## 4.1 SCORE ANALYSIS

Figure 1 shows a consistent regional gap across all models: scores in Latin America are systematically lower than in Europe and the United States for every cultural category. GPT-3.5 and GPT-5 remain the strongest systems overall, but both still experience noticeable drops in LATAM—especially for public figures, objects, and traditions. In contrast, Mistral exhibits an interesting deviation from this pattern. As an EU-trained model, it performs comparatively well on several Latin American categories, particularly dish and flora, where its LATAM scores (55.7% and 66.5%) are close to—or even higher than—its Non-LATAM scores. This suggests that some open models may generalize better than expected in specific cultural areas, even as the broader regional disparity remains pronounced.

## 4.2 PROBE EVALUATION

Beyond direct graph-based scores, we evaluate probe models trained on entity embeddings to predict the same four knowledge scores. Lower RMSE indicates better alignment between the probe's predictions and the reference (graph-based) scores.

**Table 3:** LLM-as-judge scores (RMSE ± std) across models and regions *(lower is better)*. Best (lowest) value per region is in **bold**.

| Model | Latin America | Europe | U.S. |
|---|---|---|---|
| GPT-3.5 large | **0.271 ± 0.010** | 0.233 ± 0.007 | 0.222 ± 0.004 |
| GPT-3.5 small | 0.274 ± 0.012 | 0.231 ± 0.005 | 0.223 ± 0.003 |
| GPT-5 large | 0.278 ± 0.009 | 0.235 ± 0.003 | 0.234 ± 0.002 |
| GPT-5 small | 0.282 ± 0.010 | 0.237 ± 0.005 | 0.233 ± 0.004 |
| Mistral 3.1 (24B) | 0.272 ± 0.009 | **0.222 ± 0.009** | **0.217 ± 0.005** |
| DeepSeek 7B | 0.314 ± 0.034 | 0.250 ± 0.030 | 0.257 ± 0.047 |
| Qwen2.5-7B-Instruct | 0.331 ± 0.018 | 0.295 ± 0.004 | 0.288 ± 0.009 |

Table 3 presents the LLM-as-judge errors across models and regions. The pattern is clear: all models perform worse in Latin America, with higher RMSE than in Europe or the United States. For instance, GPT-5-large particularly scores 0.234 in the U.S. and 0.235 in Europe, but increases to 0.278 in Latin America. GPT-3.5 large shows a similar gap (0.222 in the U.S., 0.233 in Europe, 0.271 in LATAM). The disparity becomes larger for smaller models: Qwen2.5-7B-Instruct reaches 0.331 in Latin America, compared with 0.295 in Europe and 0.288 in the U.S.

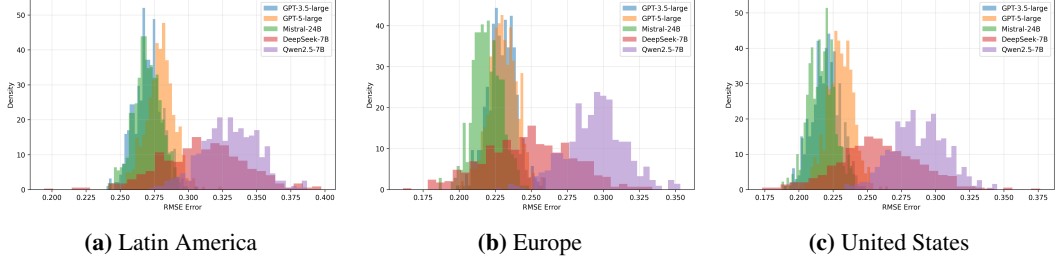

| **(a)** Latin America | **(b)** Europe | **(c)** United States |
|---|---|---|

**Figure 7:** Error distributions (RMSE) per model and region using the LLM-as-judge metric. Lower is better.

The histograms in Figures 7 show the error distributions for all models using the LLM-as-judge metric as probe supervision. Across the three regions, the pattern is consistent with the averages in Table 3, where all models exhibit a clear right-shift (higher errors) in Latin America compared with Europe and the United States. GPT-5 and GPT-3.5 display the narrowest and lowest-error distributions in all regions, while Qwen and DeepSeek present wider, heavier-tailed distributions, especially in Latin America. The separation between distributions is visually largest in the LATAM plots, confirming that the regional gap is not an artifact of averaging but is reflected across the entire error spectrum.

## 5 CONCLUSIONS

In this work, we introduce an entity-oriented methodology to evaluate how LLMs capture knowledge about Latin American entities, integrating question generation from knowledge graphs and, most importantly, a probing method extended to cultural and regional contexts. This innovation leverages question–answer scores as supervision signals to train a classifier that can directly estimate, from embeddings, the amount of knowledge about a given entity encoded in the model.

Our results consistently show that LLMs perform significantly worse on Latin America than on Europe and the United States, with the largest gaps observed in socially grounded categories such as public figures and objects, while natural domains like fauna, flora, and cuisine are comparatively better represented.

The central contribution of this work is twofold: first, to provide robust empirical evidence of regional disparities in LLMs; and second, to advance a replicable methodology based on probes that enables culturally situated evaluations. This extension allows for systematic assessment of how models encode knowledge specific to underrepresented regions.

A limitation of our study is its reliance on a restricted subset of graph data, which may constrain topical diversity. Future work will focus on expanding the dataset with multilingual and domain-specific graph resources and on strengthening the qualitative component of the evaluation. In doing so, we aim to move toward a more inclusive ecosystem of benchmarks and toward more traceable evaluation methods—capable of showing not only how much LLMs know, but also how such knowledge is represented and calibrated within their internal structures.

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

# A APPENDIX

## A.1 MODEL CONFIGURATION (MLP REGRESSOR).

We model the mapping from entity embedding to score with a lightweight MLP: `Linear(input_dim, 64)` → `BatchNorm1d(64)` → `ReLU` → `Dropout(0.3)` → `Linear(64, 1)` → `Sigmoid`. The sigmoid constrains outputs to $[0, 1]$, matching the normalized targets. The input is the embedding obtained from the prompt *"Tell me everything you know about `<entity>`"*; the target is a knowledge score (one of the four metrics or their average). We evaluate the regressor on held-out (previously unseen) entities to assess the LLM's ability to generalize across regions and categories.

## A.2 DATASET PARTITION

**Table 4:** Entity distribution across dataset splits. Each row shows the number of entities per region divided into train, validation, and test sets.

| Region | Train | Validation | Test | Total |
|---|---|---|---|---|
| Latin America | 11,401 | 1,425 | 1,425 | 14,251 |
| Europe | 12,963 | 1,620 | 1,621 | 16,204 |
| U.S. | 11,362 | 1,420 | 1,420 | 14,202 |
| **Total** | 35,726 | 4,465 | 4,466 | 44,657 |

## A.3 CATEGORY-BASED PROBE

**Table 5:** RMSE (± std) results by model, region, and category (*lower is better*). Best (lowest) value per regions and categories are in **bold**. We use as supervision the average score.

| Model | Region | Dish | Fauna | Flora | Geography | Object | Public Figure |
|---|---|---|---|---|---|---|---|
| GPT-3.5 large | Latin America | $0.226 \pm 0.011$ | $0.213 \pm 0.014$ | $0.203 \pm 0.020$ | $0.235 \pm 0.019$ | $0.250 \pm 0.023$ | $0.281 \pm 0.020$ |
| | Europe | $0.222 \pm 0.013$ | $0.209 \pm 0.009$ | $0.188 \pm 0.009$ | $0.189 \pm 0.008$ | $0.196 \pm 0.011$ | $0.202 \pm 0.007$ |
| | U.S. | $0.220 \pm 0.015$ | $0.193 \pm 0.010$ | $0.173 \pm 0.009$ | $0.177 \pm 0.007$ | $0.200 \pm 0.003$ | $0.205 \pm 0.005$ |
| GPT-3.5 small | Latin America | $0.233 \pm 0.022$ | $0.214 \pm 0.004$ | $0.202 \pm 0.009$ | $0.259 \pm 0.009$ | $0.249 \pm 0.022$ | $0.287 \pm 0.012$ |
| | Europe | $0.212 \pm 0.011$ | $0.203 \pm 0.006$ | $0.193 \pm 0.010$ | $0.193 \pm 0.008$ | $0.200 \pm 0.006$ | $0.210 \pm 0.010$ |
| | U.S. | $0.217 \pm 0.019$ | $0.199 \pm 0.013$ | $0.176 \pm 0.011$ | $0.187 \pm 0.010$ | $0.203 \pm 0.015$ | $0.208 \pm 0.009$ |
| GPT-5 large | Latin America | $0.249 \pm 0.003$ | $0.213 \pm 0.012$ | $0.194 \pm 0.013$ | $0.239 \pm 0.014$ | $0.250 \pm 0.011$ | $0.295 \pm 0.015$ |
| | Europe | $0.216 \pm 0.006$ | $0.198 \pm 0.007$ | $0.195 \pm 0.008$ | $0.199 \pm 0.008$ | $0.209 \pm 0.005$ | $0.208 \pm 0.010$ |
| | U.S. | $0.210 \pm 0.011$ | $0.192 \pm 0.004$ | $0.180 \pm 0.007$ | $0.210 \pm 0.005$ | $0.209 \pm 0.006$ | $0.220 \pm 0.007$ |
| GPT-5 small | Latin America | $0.235 \pm 0.017$ | $0.224 \pm 0.013$ | $0.194 \pm 0.008$ | $0.237 \pm 0.016$ | $0.244 \pm 0.017$ | $0.297 \pm 0.006$ |
| | Europe | $0.226 \pm 0.008$ | $0.197 \pm 0.006$ | $0.192 \pm 0.003$ | $0.198 \pm 0.009$ | $0.208 \pm 0.003$ | $0.206 \pm 0.003$ |
| | U.S. | $0.214 \pm 0.017$ | $0.192 \pm 0.004$ | $0.178 \pm 0.007$ | $0.216 \pm 0.015$ | $0.205 \pm 0.007$ | $0.214 \pm 0.007$ |
| MISTRAL 3.1-small (24B) | Latin America | $0.238 \pm 0.011$ | $0.198 \pm 0.014$ | $0.192 \pm 0.006$ | $0.235 \pm 0.015$ | $0.228 \pm 0.012$ | $0.269 \pm 0.009$ |
| | Europe | $0.208 \pm 0.006$ | $0.186 \pm 0.007$ | $0.183 \pm 0.008$ | $0.191 \pm 0.005$ | $0.192 \pm 0.008$ | $0.204 \pm 0.010$ |
| | U.S. | $0.218 \pm 0.012$ | $0.184 \pm 0.008$ | $0.179 \pm 0.006$ | $0.189 \pm 0.008$ | $0.210 \pm 0.008$ | $0.213 \pm 0.008$ |
| DeepSeek 7b | Latin America | $0.258 \pm 0.031$ | $0.278 \pm 0.034$ | $0.216 \pm 0.044$ | $0.253 \pm 0.031$ | $0.263 \pm 0.026$ | $0.317 \pm 0.034$ |
| | Europe | $0.230 \pm 0.039$ | $0.262 \pm 0.046$ | $0.221 \pm 0.050$ | $0.224 \pm 0.033$ | $0.237 \pm 0.039$ | $0.223 \pm 0.052$ |
| | U.S. | $0.250 \pm 0.043$ | $0.244 \pm 0.033$ | $0.240 \pm 0.037$ | $0.229 \pm 0.053$ | $0.297 \pm 0.020$ | $0.220 \pm 0.059$ |
| QWEN | Latin America | $0.266 \pm 0.017$ | $0.306 \pm 0.016$ | $0.274 \pm 0.009$ | $0.273 \pm 0.019$ | $0.261 \pm 0.019$ | $0.300 \pm 0.012$ |
| | Europe | $0.259 \pm 0.014$ | $0.290 \pm 0.010$ | $0.270 \pm 0.017$ | $0.249 \pm 0.017$ | $0.266 \pm 0.009$ | $0.271 \pm 0.012$ |
| | U.S. | $0.252 \pm 0.015$ | $0.271 \pm 0.021$ | $0.243 \pm 0.008$ | $0.261 \pm 0.012$ | $0.265 \pm 0.010$ | $0.268 \pm 0.015$ |

## A.4 EXAMPLE OF QUESTIONS AND ANSWERS

**Table 6:** Example of entity-based evaluation for the triplet `[Neruda, profession, poet]`. Models are asked: *"What is the profession of Pablo Neruda?"*.

| Ground truth | Poet |
|---|---|
| **GPT-3.5** | Answer: "Poet" |
| **GPT-5** | Answer: "Poet, one of the most famous Chilean poets" |
| **QWEN** | Answer: "Poet and politician" |
| **Mistral** | Answer: "Poet" |
| **DeepSeek** | Answer: "Writer, diplomat, and poet" |
| **Score (LLM-as-judge)** | 1.0 |

**Table 7:** Example of entity-based evaluation for the triplet `[['Capón', 'origen', 'Argentina y Uruguay']`. Models are asked: *"What is the origin country of the capon?"*.

| Ground truth | Argentina and Uruguay |
|---|---|
| GPT-3.5 | Answer: "The capon originated in France." |
| GPT-5 | Answer: "It comes from the castration of a young rooster to fatten it and improve the quality of its meat." |
| QWEN | Answer: "The origin of the capon is the capon, which is an animal known for its ability to breathe and breathe in a respiratory way." |
| Mistral | Answer: "The capon is a castrated rooster bred to improve its meat." |
| DeepSeek | Answer: "The origin of the capón is the sheep breed called "Cabrito de montaña" or "Iberian mountain kid" in Spain." |
| Score (LLM-as-judge) | 0.2 |