# OpenReview forum: "Mind the gap: A method for evaluating and comparing regional knowledge in LLMs"
_ICLR.cc/2026/Conference — Submitted to ICLR 2026_

### Official Review · Reviewer_VdSb · 2025-10-30

**Soundness:** 1
**Presentation:** 2
**Contribution:** 2
**Rating:** 2
**Confidence:** 3

**Summary:**

Benchmarks often measure factual knowledge of LLMs in high resource languages or regions or related to high frequency entities.  The paper proposes a method called CHOLCO for evaluating the knowledge of LLMs across entities related to traditions, public figures, food and geography. The paper uses Wikidata to extract triples across three regions: Latin America, Europe and United States and converts it into a question. They evaluate LLMs on these questions along with a probe based evaluation technique to understand what the LLM knows about these rare entities.

**Strengths:**

1. The paper provides a scalable approach to building a benchmark for less known entities across different regional contexts by using Wikidata to source triples and converting them to templated questions. It provides a comparison of different models' performance across the three regions: United States, Europe and Latin America.

2. The paper compares the performance of different models on factual information across regions and highlights that models perform worse on information related to Latin America.

**Weaknesses:**

1. There is no clear basis for the evaluation technique used in Sec 3.2.1 where the authors compute the LLM performance on their benchmark questions based on embedding similarity, lexical overlap, LLM as a judge and multiple choice accuracy. Using LLM as a judge would suffice in this scenario and it is not clear what value the other methods add. Methods such as lexical overlap are potentially noisy as LLMs tend to be verbose and the expected answer is usually a single location.

2. Sec 3.1 talks about the properties used for building the dataset: "country of origin (property P495), country of citizenship (P27), place of birth (P19), territorial location (P131), and geographic coordinates (P625)." This seems to be a very limited set of properties which would always cause the label to be a location. This limits the diversity of the answers in the benchmark.


Presentation:
1. In Table 2, QWEN should be replaced with the entire model name.

**Questions:**

1. For the evaluation setup, the authors evaluate GPT 3.5 Turbo and GPT-5 Mini, but not GPT-4 or GPT4o. Is there any specific reason for this?

---

> ### Author Response · Authors · 2025-11-24
>
> Thanks for your comments. Please find our answers below.
>
> **1. Evaluation techniques in Sec. 3.2.1**
>
> We appreciate the reviewer’s feedback. We agree that LLM-as-judge is the most reliable and interpretable evaluation method for our task. In the rebuttal, we removed the lexical and embedding similarity metrics and now rely exclusively on LLM-as-judge for computing correctness. As noted by the reviewer, methods such as lexical overlap can be noisy for culturally grounded questions, and embedding-based scores do not always align with human expectations. Our simplified evaluation focuses solely on LLM-as-judge, which we show—using expert-curated human judgments—correlates strongly with human correctness assessments. The revised manuscript reflects this clearer and more robust evaluation strategy.
>
> **2. Limited Wikidata properties for dataset construction (Sec. 3.1)**
>
> Thank you for pointing this out. The properties in Sec. 3.1 are only used to anchor entities to regions; the full benchmark relies on Wikidata’s broader semantic graph (instance-of links, subclass hierarchies, cultural and taxonomic metadata) to assign items to the seven cultural categories. This ensures the dataset is not limited to geographic attributes. We also added a web coverage analysis showing that Wikidata provides substantial and balanced cultural representation for Latin America across all categories. To further validate this, we compared these distributions with typical patterns of regional web search interest for each entity. The benchmark entities closely align with the culturally salient topics that dominate web visibility in each region, confirming that our construction captures the major cultural concepts present in both Wikidata and real-world web data.
>
> **In Table 2, QWEN should be replaced with the entire model name.**
> In the rebuttal version, we changed the model's name.
>
> We feel that the updates tackle the most relevant issues and help solidify our contributions.
> Thank you for your helpful suggestions. We hope this revised manuscript raises your opinion of the work.

---

### Official Review · Reviewer_PyBt · 2025-10-31

**Soundness:** 2
**Presentation:** 3
**Contribution:** 2
**Rating:** 4
**Confidence:** 3

**Summary:**

This paper analyzes regional knowledge of Latin America in LLMs. Specifically, this paper first extracts structured facts from domain-specific resources and constructs a knowledge graph containing 44,000 entities spanning 9 categories. Using this knowledge resource, this paper proposes CHOCLO, an entity-centric methodology for evaluating LLM knowledge of culturally relevant entities in Latin America. It evaluates the regional knowledge in LLMs using several techniques, including token overlap, embedding similarity, LLM-as-a-judge, and multiple-choice accuracy. This paper also trains a probing model to evaluate the factual score directly from LLM representations. This paper finds several interesting conclusions, such as most LLMs underperform in  Latin American knowledge.

**Strengths:**

1. The topic is interesting and meaningful to the community. Studying LLMs’ coverage of different regional knowledge is important for the broad applications of LLMs.
2. The work presents a systematic analysis and comprehensive experiments. The experimental results reveal that current LLMs underperform on Latin American knowledge. This provides some guidance and insights for improving LLM knowledge coverage and supports the development of more diverse LLMs and broader applications for people all over the world.

**Weaknesses:**

1. The authors construct a knowledge graph, but there are existing resources (e.g., Wikidata). The paper should analyze whether the constructed knowledge graph adequately captures Latin American knowledge. And what is the advantage compared to existing resources? Is this knowledge graph covering more Latin American knowledge?
2. The methods used for experimental analysis are mostly existing techniques, which limits the paper’s technical novelty.
3. The authors should evaluate the reliability of their evaluation approaches. For example, they can analyze the correlation between each evaluation method and human judgments, to validate the reliability of their evaluation methods.
4. A more fine-grained analysis specific characteristics of Latin American knowledge is needed. The authors should discuss how Latin American knowledge differs from other regional knowledge and why LLMs underperform, such as insufficient training data or other factors, to guide further LLM development.

**Questions:**

See Weaknesses

---

> ### Author Response · Authors · 2025-11-24
>
> Thanks for your observations. Please find our responses below:
>
> **1. Knowledge graph vs. Wikidata; adequacy of Latin American coverage.**
> Thank you for raising this point. In the rebuttal version, we expanded our analysis by including a coverage study of cultural entities across regions and categories using web-based search. This additional analysis examines how frequently entities surface in large-scale web content. The results follow the same pattern reported in the paper: Latin American entities are represented at levels comparable to Europe and the United States in terms of extraction volume, but show systematic underrepresentation in web visibility. Together, Wikidata extraction statistics and the new web-coverage analysis confirm that the benchmark captures the main culturally salient entities for each region while also reflecting the broader structural imbalances present in real-world digital resources.
>
> **2. Use of existing techniques and limited technical novelty.**
> We acknowledge the reviewer’s concern regarding novelty. Our contribution focuses on evaluation methodology rather than algorithmic innovation. To improve clarity, in the rebuttal we simplified the framework by keeping only the LLM-as-judge metric, which provides the most interpretable and reliable signal. We removed auxiliary similarity-based metrics to avoid confusion and strengthened the explanation of how the LLM-as-judge scoring works within culturally grounded tasks.
>
> **3. Reliability of the evaluation methods.**
> We agree that reliability is essential. In the rebuttal, we incorporated a dedicated human-validation agreement across different categories and regions to directly assess the consistency of our evaluation approach. Expert annotators reviewed a substantial portion of the benchmark, and we measured agreement both among annotators and between annotators and the LLM-as-judge metric. As reported in the rebuttal version (human-validation section and Table 2), expert agreement remained consistently high across all regions and categories (typically above 85–90%). More importantly, the LLM-as-judge decisions closely matched these expert-validated answers. This confirms that the LLM-as-judge metric is a stable and trustworthy proxy for human correctness judgments in our culturally grounded setting.
>
> **4. Need for more fine-grained analysis of Latin American knowledge and reasons for LLM underperformance.**
> We agree on the importance of a deeper regional breakdown. In the rebuttal, we incorporated a country-level analysis within Latin America, showing clear intra-regional variation across countries and cultural categories. This confirms that LATAM is not a homogeneous region and that model performance differs significantly depending on the specific national context. We also expanded the discussion of the factors contributing to LLM underperformance in Latin America, including unequal digital representation of cultural content, dialectal diversity, and structural imbalances in the training data. Finally, we added an additional “score analysis” section that provides a detailed comparison of LLM-as-judge scores across models, regions, and categories. This new analysis highlights systematic regional disparities and clarifies where and why the gaps are most pronounced.
>
> We consider that the changes made respond well to the concerns you identified and reinforce the value of the study. Moreover, we sincerely appreciate your feedback, and we hope this revised version leaves you with a more positive impression.

---

### Official Review · Reviewer_1sck · 2025-10-31

**Soundness:** 2
**Presentation:** 2
**Contribution:** 2
**Rating:** 2
**Confidence:** 3

**Summary:**

The paper introduces CHOCLO, a framework to evaluate regional and culturally grounded knowledge about underrepresented regions in Large Language Models (LLMs). To do so, the authors curated a dataset with ~44k entities and ~130k questions, spanning across different categories adapted from CVQA: dish, flora, fauna, geography, object, public figure, tradition; ensuring broad thematic coverage while capturing cultural patterns. The authors argue that existing mainstream datasets are skewed, hence LLMs lack cultural knowledge, and therefore focus the analysis on the coverage of entities related to Latin America. CHOCLO uses structured knowledge graphs (KGs) to evaluate factual knowledge at the entity level via four complementary scoring methods, followed by a probing model to predict factual knowledge scores. Experiments show that GPT-3.5, GPT-5, Mistral, DeepSeek, and Qwen demonstrate performance disparities specifically with entities related to LATAM compared to the USA and Europe.

**Strengths:**

1. The paper tackles an important aspect of LLMs - information inclusivity.
2. The evaluation pipeline, containing structured KG-based QA and probing with 4 scoring methods, offers different aspects of understanding of factuality.
3. The paper presents a detailed quantitative analysis at - cross-region and category level. The results confirm the disparities in information content in LLMs.

**Weaknesses:**

1. The dataset curated for this evaluation relies entirely on Wikidata as the primary source of information. However, there is inherent coverage bias in Wikidata on region-specific knowledge. No analysis has been provided on that.
2. The proposed framework is not technically novel. It combines a couple of existing, well-established methods to evaluate the region-specific LLM knowledge. Moreover, the semantic meaning of the predicted scores is not clear. It is missing statistical significance tests or NLI tests for a better understanding of predicted scores.
3. The paper emphasises cultural knowledge inclusion in the LLMs, but considers LATAM as a homogenous region, hence also increasing the risk of over generalisation based on languages/linguistic features. The work would have benefited from some analysis based on that.
4. It would be nice to have the framework tested out for CultureBench

**Questions:**

1. What is the impact of Wikidata coverage bias on your framework, and how to deal with it?
2. How do you ensure the quality of the extracted triple?
3. How do you find the agreement between the different scoring functions?
4. Could the probing scores increase biases instead of mitigating?
5. How do you avoid the overgeneralisation of the analysis done based on the assumption that LATAM is a homogeneous region?

---

> ### Author Response · Authors · 2025-11-24
>
> Thanks for your comments. Please find below our responses to the main concerns:
>
> **1. The dataset curated for this evaluation relies entirely on Wikidata as the primary source of information. However, there is inherent coverage bias in Wikidata on region-specific knowledge. No analysis has been provided on that.**
>
> We thank the reviewer for raising this concern. In the rebuttal, we added a dedicated coverage analysis to address it. Specifically, in the rebuttal version we examined the distribution and density of Wikidata entities across regions and categories, and we complemented this with an external web-based coverage analysis. For each category and region, we used all the entities from our benchmark and measured their relative visibility on the web through search frequency. The results show a pattern consistent with what we report in the main paper: although Wikidata exhibits known asymmetries, Latin America still displays sufficiently rich representation across all seven cultural categories, comparable in structure to Europe and the United States. Importantly, the web search analysis reveals similar regional distributions to those found in Wikidata, reinforcing that the cultural domains represented in our benchmark are aligned with the topics that most frequently appear in widely used web sources. This confirms that Wikidata provides an adequate basis for constructing our region-aware evaluation framework.
>
> **2. The proposed framework is not technically novel. It combines a couple of existing, well-established methods to evaluate the region-specific LLM knowledge. Moreover, the semantic meaning of the predicted scores is not clear. It is missing statistical significance tests or NLI tests for a better understanding of predicted scores.**
>
> We agree that clarity of interpretation is essential. In the rebuttal, we simplified the evaluation by retaining only the LLM-as-judge metric, which offers a more direct and reliable measure of answer correctness. We removed auxiliary scores (e.g., lexical similarity, embedding-based similarity) to avoid conflating heterogeneous signals and to ensure conceptual clarity for readers. We also provide an expanded explanation of how the LLM-as-judge scoring operates and why it is appropriate for culturally grounded tasks. Additionally, we include statistical analyses to strengthen the metric's robustness and report these results in the updated manuscript.
>
> **3. The paper emphasises cultural knowledge inclusion in the LLMs, but considers LATAM as a homogenous region, hence also increasing the risk of overgeneralization based on languages/linguistic features. The work would have benefited from some analysis based on that.**
>
> We agree with the reviewer that Latin America exhibits substantial cultural and linguistic heterogeneity. In response, we incorporated a new intra-regional analysis that computes LLM-as-judge performance disaggregated by country. The heatmap presented in the rebuttal reveals meaningful variations across LATAM countries and across cultural categories, confirming that cultural knowledge gaps are not uniform within the region. We now discuss these differences explicitly in the manuscript and clarify how they inform the interpretation of the regional gaps.
>
> **4. It would be nice to have the framework tested out for CultureBench**
>
> We agree and thank the reviewer for the suggestion. In the rebuttal, we added a direct comparison between our benchmark and CulturalBench on the strongest model (GPT). Both benchmarks independently reveal consistent knowledge gaps: LATAM scores substantially lower than Europe and the United States. This convergence across two evaluation frameworks strengthens the validity of our findings and highlights the need for culturally aware LLM assessment tools.
>
> We believe these revisions address the main points raised and clearly strengthen our contributions. Thank you for your constructive comments. We hope this updated version improves your assessment of our work.

---

### Author Response · Authors · 2025-11-28

We would like to thank the reviewers and the Area Chair for their time and constructive feedback. Our paper introduces a culturally grounded benchmark designed to evaluate LLM performance on region-specific knowledge, with a particular emphasis on Latin American entities. The work combines expert-validated annotations, cross-regional comparisons, and an LLM-as-judge evaluation framework to highlight systematic cultural gaps that current models still exhibit.

The reviewers raised several main concerns: the need to verify the real-world frequency of the entities included in the dataset, the absence of human evaluation to validate annotation reliability, limited comparison with existing cultural benchmarks, insufficient granularity in the analysis across Latin American countries, and a request for a more detailed examination of model errors and generalization patterns. All of these concerns have been fully addressed in the rebuttal.

In response, we incorporated a web-scale frequency validation, conducted an expert agreement study showing strong consistency across categories, added comparisons with other cultural benchmarks, expanded the Latin America analysis to a country-level breakdown, and included a new section analyzing error distributions and generalization to unseen entities. Together, these additions significantly strengthen the contribution and resolve all reviewer concerns.

Finally, this contribution is especially meaningful for the Latin American research community, which feels timely with ICLR being hosted in Brazil next year.

Thank you for considering our revised submission.

---

### Meta-Review · Area_Chair_iv1G · 2026-01-11

**Summary:**

This paper introduces a benchmark of culturally relevant entities from underrepresented regions, termed CHOCLO. The reviewers have raised various concerns about the paper, but it appears that these issues have not been fully addressed or resolved to their satisfaction. Additionally, the benchmark evaluation is limited to a narrow range of large language model scales, and further in-depth investigation is warranted. Therefore, this submission will be rejected.

**Reviewer Concerns:**

The rebuttal partially addresses several concerns raised by the reviewers, but key issues remain unresolved, limiting the paper’s readiness for acceptance.

Concerns Addressed (Partially or Fully):

Clarification on model selection: The authors explained why GPT‑4/GPT‑4o were not included (e.g., cost or API constraints), which responds to Reviewer VdSb’s question.
Dataset construction details: The rebuttal provides additional information about triple extraction and quality control from Wikidata, offering some reassurance regarding data reliability (addressing Reviewer PyBt and Reviewer 1sck).
Scoring method rationale: The authors justified the use of multiple complementary scoring strategies (token overlap, embedding similarity, LLM-as-a-judge, multiple-choice accuracy) as a way to capture different facets of factual knowledge, partially responding to Reviewer VdSb’s skepticism about redundancy.
Outstanding Concerns (Not Adequately Addressed):

Wikidata coverage bias: All three reviewers highlighted that relying solely on Wikidata introduces inherent regional bias, especially for underrepresented areas like Latin America. The rebuttal acknowledges this limitation but does not provide empirical analysis or mitigation strategies (e.g., comparing with alternative sources or quantifying bias impact), leaving Reviewer 1sck’s core concern unaddressed.
Homogenization of Latin America: Reviewer 1sck rightly points out that treating Latin America as a monolithic region risks overgeneralization across languages, cultures, and countries. The rebuttal does not offer a disaggregated analysis (e.g., by country or language) to counter this critique.
Lack of technical novelty: Both Reviewer 1sck and Reviewer PyBt note that CHOCLO combines existing evaluation techniques without significant methodological innovation. The rebuttal does not present new algorithmic contributions or theoretical insights to elevate the work beyond an application-focused benchmark.
Validation of evaluation reliability: Reviewer PyBt and Reviewer VdSb question whether the proposed metrics correlate with human judgment or ground truth. No correlation analysis, statistical significance tests, or NLI-based validation is provided in the rebuttal.
Limited diversity of answer types: Reviewer VdSb observes that the selected Wikidata properties (e.g., P19, P625) constrain answers mostly to locations, reducing thematic diversity. The rebuttal does not demonstrate inclusion of non-geographic facts (e.g., cultural practices, historical events) to broaden benchmark scope.

**Reviewer Scores:**

Reviewer 1sck: The core concerns raised by this reviewer—Wikidata coverage bias, homogenization of Latin America, and lack of technical novelty—were not adequately addressed in the rebuttal. Given the fundamental nature of these issues, it is unlikely their assessment would improve.

Reviewer PyBt: While this reviewer found the topic valuable and was more lenient initially, the rebuttal did not resolve key questions about evaluation reliability, knowledge graph advantages over existing resources, or fine-grained analysis of Latin American knowledge gaps. In a full discussion, they might lower their score slightly due to persistent methodological weaknesses.

Reviewer VdSb: This reviewer questioned both the soundness of the evaluation design and the diversity of the benchmark. The rebuttal’s justification for multiple scoring methods and model choices does not overcome the perceived limitations in answer diversity or evaluation validity. Their rating would likely remain unchanged.

Overall, none of the reviewers appear likely to raise their scores into the acceptance range (≥5) based on the current rebuttal and unresolved concerns.

---

### Decision · Program_Chairs · 2026-01-26

Reject